# Mid-life social participation in people with intellectual disability: The 1958 British birth cohort study

Zuyu Wang[1], Andrew Sommerlad[1,2], Angela Hassiotis[1,2], Marcus Richards[3], Gill Livingston[1,2]*

1 Division of Psychiatry, University College London, London, United Kingdom, 2 Camden and Islington NHS Foundation Trust, London, United Kingdom, 3 MRC Unit for Lifelong Health & Ageing at UCL, University College London, London, United Kingdom

* g.livingston@ucl.ac.uk

**Data Availability Statement:** All relevant data are within the paper and its Supporting Information files.

## Abstract

### Background

Low social participation is a potentially modifiable risk factor for cognitive deterioration in the general population and related to lower quality of life (QoL). We aimed to find out whether social participation is linked to cognitive deterioration and QoL for people with borderline intellectual functioning and mild intellectual disability.

### Method

We used data from the National Child Development Study, consisting of people born during one week in 1958, to compare midlife social participation in people with mild intellectual disability, borderline intellectual functioning, and without intellectual impairment. We defined social participation as 1. confiding/emotional support from the closest person and social network contact frequency at age 44, and 2. confiding relationships with anyone at age 50. We then assessed the extent to which social participation mediated the association between childhood intellectual functioning and cognition and QoL at age 50.

### Results

14,094 participants completed cognitive tests at age 11. People with borderline intellectual functioning and mild intellectual disability had more social contact with relatives and confiding/emotional support from their closest person, but fewer social contacts with friends and confiding relationships with anyone than those without intellectual disability. Having a confiding relationship partially mediated the association at age 50 between IQ and cognition (6.4%) and QoL (27.4%) for people with borderline intellectual functioning.

### Conclusion

We found adults with intellectual disability have positive family relationships but fewer other relationships. Even at the age of 50, confiding relationships may protect cognition for people with borderline intellectual functioning and are important for QoL.

**Funding:** MR is funded by the Medical Research Council (MC_UU_00019/1 and 3). The funders had no role in study design, data collection and analysis, decision to publish, or preparation of the manuscript.

**Competing interests:** The authors have declared that no competing interests exist.

## Background

Social participation is an important human need and linked to health outcomes including improved cognitive function [1, 2]. However, adults with borderline intellectual functioning (defined here as IQ <85 but ≥70), and intellectual disability (IQ <70) [3] may experience difficulties in building and maintaining desired social relationships. Communication difficulties [4], societal stigma, and cognition-related limitations to an independent social life [5–7] may impede their social participation. Studies show that individuals with intellectual disability typically have more restricted social networks than the general older population [8] and primarily interact with their families and caregivers [9, 10]. However, these studies have been cross-sectional, usually unrepresentative of the population of people with intellectual disability or have not compared people with intellectual functioning or disability to the general population. Thus, findings may not be generalisable.

Social participation is associated with better quality of life (QoL) for adults with intellectual disability [11] and with cognition in the general population [1, 12, 13]. However, people who decline cognitively in later life may be less interested in or less able to engage in or arrange social participation. This suggests a circular relationship whereby poor social participation worsens cognition, which further worsens social participation and QoL. Social participation may therefore be an important and modifiable mediator of the relationship between intellectual disability and cognition and QoL, but no previous studies have examined this.

This study therefore aims to answer the following questions: 1) Does social participation differ between people with and without borderline intellectual functioning/disability in midlife (defined as 44 to 50 years old)? 2) Does social participation (social contacts and confiding relationships) mediate the association between intelligence quotient scores (IQ) in childhood and future cognition and QoL in people with and without intellectual disability?

## Method

### Study design and participants

We used data from the National Child Development Study (NCDS) [14]. This is a longitudinal cohort initially formed of 17,415 people born in one week of March 1958 in England, Scotland, and Wales. It later added 1143 people born that week who immigrated to this country. Since the initial birth sweep in 1958, participants have been asked to take part in ten follow-up assessments via questionnaires with face-to-face clinic assessments. Study participants for our analysis are cohort members who completed all baseline cognition tests at age 11 to enable us to calculate an IQ score.

### Measures

**Demographic information.** We obtained participants' sex; social class defined as the employment of the mother's partner at birth using The Registrar-General's Social Class Scheme: The Stevenson Version [15] (RG I, II, III non-manual, III manual, IV, V); ethnic groups (White/White other, Black/Black British, Asian/Asian British, Mixed race, other or unsure) from responses at ages 7 and 16; legal marital status was defined at age 44 (Single/ Never married, Married, separated/divorced/widowed).

**IQ in childhood.** Participants completed cognitive tests at ages 7 and 11. We used test results at age 11 when the widest range of the tests were completed to calculate childhood IQ. Participants completed four tests at this age: General Ability (80 Verbal and Non-Verbal multiple-choice questions), Reading Comprehension test (completing 35 sentences), Mathematics test (number skills, fractions, measures, and geometry), and Copying Designs (copying 6

drawings), all tests were performed by schoolteachers. We calculated completers' childhood IQ by conducting a Principal Component Analysis (PCA) on these scores (N = 14,094), to obtain component scores, which was also done in a previous study [16]. We transformed the resulting scores to a scale with a mean of 100 (Standard deviation [SD] = 15) to correspond to IQ scores (min = 50.1, max = 139.6). These scores were normally distributed. We categorised participants into three groups ">85", "borderline intellectual functioning (IQ 70–85)", and "mild intellectual disability (50<IQ<70)" based on the standard intellectual disability definition [3].

**Social participation: Social contact with relatives and friends.** 'Social participation' has been conceptualised in a variety of ways with different terms to describe social domains, e.g., connections and engagement [17]. In this study, we define social participation as direct participation in interactions with others including confiding relationships, in line with a recent consensus definition [18]. At age 44 participants completed questionnaires about their social life, using six questions from the Berkman Syme social network index [1, 19] (Table A in S1 Appendix). These measures previously showed an association with subsequent dementia and declining cognition in a study of UK civil servants [1]. Each question is answered on a Likert-scaled for the number or frequency of people seen, with a higher score indicating more social contact. We generated a 'relatives' subscale from questions 1 to 3, a 'friends' subscale from questions 4 to 6, and a total social contact score by summing all 6 scores. The possible total social contact index is 30, with 15 each for relatives or friends.

**Social participation: Confiding/emotional support from the closest person.** At age 44, participants completed the Close Person Questionnaire (CPQ) [20], which assesses emotional and practical social support from one close person nominated by the respondent, either "husband/wife/partner", "boyfriend/girlfriend", "Parent", "Siblings"(brother/sister), "Children" (son/daughter), "Other relatives", "Neighbour", "Friend from work", "Other friends" and "Others". We used the Confiding/emotional support subscale which has eight questions (possible scores from 0 to 24) and asks the respondent to consider the emotional support from the person they felt closest to in the preceding 12 months (Table B in S1 Appendix). All questions have four response options: Not at all = 0, A little = 1, Quite a lot = 2, A great deal = 3. Higher scores correspond to a more confiding relationship or higher social support.

**Social participation: Other confiding relationships.** At age 50, the participants were asked "If you needed to talk about your problems and private feelings how much would the people around you be willing to listen?". This question has four response options: Not at all = 0, A little = 1, Somewhat = 2, A great deal = 3.

**Measurement of cognitive ability in midlife.** Cognition was tested at age 50. We performed PCA on three tests: memory (word list recall), speed of processing (letter cancellation) and verbal fluency (animal naming) [21] which were completed at age 50, then standardised test scores to z-score (mean = 0, SD = 1).

**Quality of life.** QoL was measured at age 50 using the 12-item Control, Autonomy, Self-realization, and Pleasure Version 2 (CASP-12 v.2) [22] which has four dimensions: control, autonomy, self-realization, and pleasure [23] (Table C in S1 Appendix). Each Likert-scaled item has four options: "Often", "Sometimes", "Not often", and "Never" (score 0 to 3). Possible total score ranges from 0 to 36, with higher scores indicating better QoL.

**Health status.** General health was rated at age 44 when participants were asked: "How would you describe your health generally" with potential responses of "Excellent", "Good", "Fair", and "Poor". Mental Health was measured using the validated Clinical Interview Schedule-Revised (CIS-R) [24] at age 44. The scores are summed to give an overall severity score from 0 to 4 for fatigue, concentration and forgetfulness, sleep problems, irritability, depression, depressive ideas, anxiety, phobias, and panic, with possible total scores ranging from 0 to 36. A

score of 12 or more indicates a significant level of symptoms, and a score of 18 or more suggests that treatment is needed.

**Statistical analysis.** We used Stata MP 17.0 for statistical analyses. We first described participants' demographic characteristics and social participation.

1. **To calculate the association between childhood IQ classifications and mid-life social participation.** We used Multiple Imputation by Chained Equations (MICE) [25] to impute missing covariate data, including ethnic group, social class, legal marital status at age 44, self-reported general health at age 44, the continuous total CIS-R scores at age 44. We then used linear regression using those with IQ above 85 as the reference group, to examine the relationships between IQ classifications in childhood and: 1) Social contact with relatives and friends at age 44, 2) confiding relationship with one person at age 44 and with anyone at age 50. The model was initially unadjusted (model 1), then adjusted for sex, ethnic group, and social class (model 2), then additionally adjusted for legal marital status at age 44 and self-reported general health at age 44 (model 3), and finally additionally adjusted for continuous CIS-R scores at age 44 (fully adjusted model- model 4).

2. **To examine whether social participation mediates the relationships between IQ in childhood and cognition, and QoL at age 50.** We used Baron and Kenny's mediation test [26] (Text in S2 Appendix). It consists of two main steps, 1) linear regression and 2) statistical significance tests (Sobel's z-test, Bootstrapping [27]). The possible results are no, partial or full mediation [26, 28].

# Results

## Description of the cohort

Fig 1 shows the flow of people through the study. At age 11, 14,094 (75.9%) participants completed all cognitive tests, 40 (0.2%) completed some tests and 4424 (23.8%) completed no tests. At age 44, 8039 (57.0%) participants with IQ test scores participated, including 6182 (43.9%) participants who completed questions about their social contact with relatives, 6650 (47.2%) participants who completed questions about social contact with friends, 7202 (51.1%) participants who completed questions about confiding/emotional support from the closest person. At age 50, 8448 (59.9%) participants with IQ scores participated and completed questions about confiding relationships with anyone.

**Participants.** We summarised the demographic characteristics of our sample in Table 1. There were 235 people with mild intellectual disability (1.7%), 2234 people with borderline intellectual functioning (15.9%) and 11625 people with IQ above 85 (normal group) (82.5%). Overall, 51.4% of participants were male, but 54.7% of those with borderline intellectual functioning and 59.2% of people with mild intellectual disability were male. At age 44, 45.3% of people with mild intellectual disability, 65.9% of people with borderline intellectual functioning and 73.1% of people with IQ above 85 were married. At age 44, 56.6% of people with mild intellectual disability, 70% of people with borderline intellectual functioning, and 82.7% of people with IQ above 85 reported their health as excellent or good.

**Social network.** The mean score of Berkman-Syme social network index for adults with mild intellectual disability was 23.5 (SD = 3.1), for those with borderline intellectual functioning was 22.6 (SD = 3.4), and for those with IQ above 85 22.6(SD = 3.1). The mean score for social contact with relatives for adults with mild intellectual disability was 11.8 (SD = 1.9), for adults with borderline intellectual functioning was 11.3 (SD = 2.1), and 11.1 (SD = 1.9) for people with IQ above 85. The mean score of social contact with friends was 11.8 (SD = 2.1) for

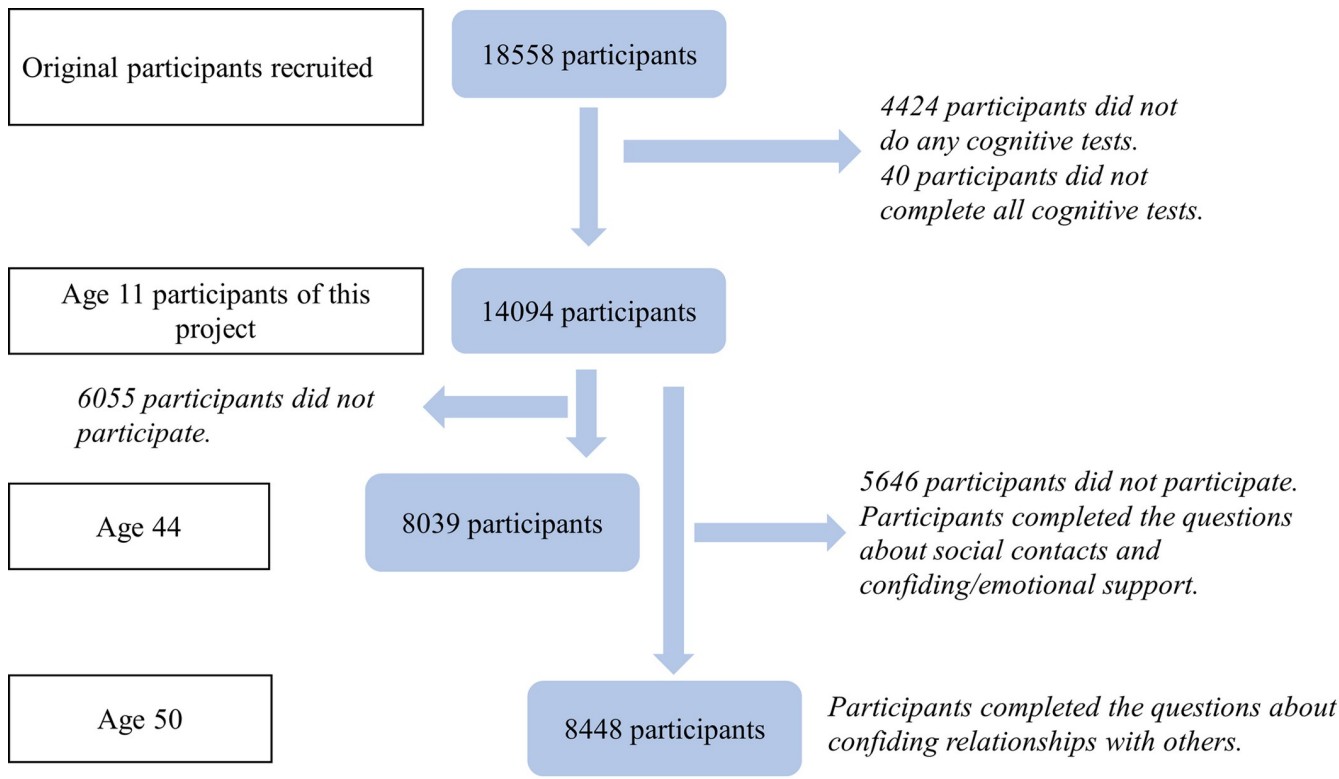

**Fig 1. Flow of participants through the study.**

adults with mild intellectual disability), 11.2 (SD = 2.2) for people with borderline intellectual functioning and 11.5 (SD = 2.2) for adults with IQ above 85.

**Confiding/emotional support.** 6792 (84.5%) participants nominated a close person at age 44. For people with mild intellectual disability, their closest relationships were husband/wife/partner (47.2%), or parents (27.8%), followed by others (8.3%), siblings and other friends (both 5.6%), boyfriend/girlfriend and neighbours (both 2.8%). Unlike the other two groups, none reported their closest person as children or other relatives or friends from work. People with borderline intellectual functioning reported their closest relationship to be a spouse or partner (71.8%), other friends (5.9%), parents (5.6%), and followed by siblings (5.1%), children (4.8%), boyfriend/girlfriend (3.9%), other relatives (1.2%), friends from work (1.1%) and neighbours (0.5%) and others (0.3%). The majority of people with IQ above 85, reported their closest relationships are with their spouse or partner (76.1%), followed by other friends (7.1%), parents (3.8%), boyfriend/girlfriend (3.7%), children (3.4%), siblings (3.3%), friends from work (1.5%), other relatives and others (both 0.4%), and neighbours (0.3%).

The mean score of confiding/emotional support from the closest person for people with mild intellectual disability was 15.0 (SD = 5.1), for people with borderline intellectual functioning was 14.7 (SD = 4.4), and for adults with IQ above 85 was 15.3 (SD = 4.0). Mean scores for confiding relationships with anyone were 3.5 (SD = 0.9) for people with mild intellectual disability, 3.5 (SD = 0.9) for people with borderline intellectual functioning, and 3.7 (SD = 0.7) for people with IQ above 85

**QoL.** The mean QoL at age 50 for people with mild intellectual disability was 23.0 (SD = 5.9); it was 25.0 (SD = 6.1) for people with borderline intellectual functioning and 26.3 (SD = 5.7) for people with IQ above 85.

**Table 1. Socio-demographic of study participants.**

| Variable name | All Participants | | People with IQ above 85 | | People with borderline intellectual functioning | | People with mild intellectual disability | |
|---|---|---|---|---|---|---|---|---|
| | N | %, Mean (SD) | N | %, Mean (SD) | N | %, Mean (SD) | N | %, Mean (SD) |
| **Sex** | N = 14094 | | N = 11625 | | N = 2234 | | N = 235 | |
| *Male* | 7239 | 51.4 | 5879 | 50.8 | 1221 | 54.7 | 139 | 59.2 |
| *Female* | 6855 | 48.6 | 5746 | 49.4 | 1013 | 45.3 | 96 | 40.9 |
| **Ethnicity** | N = 14094 | | N = 11625 | | N = 2234 | | N = 235 | |
| *White/White other* | 12025 | 97.3 | 10142 | 98.1 | 1721 | 93.8 | 162 | 89.0 |
| *Black/Black British* | 124 | 1.0 | 64 | 0.6 | 51 | 2.8 | 9 | 5.0 |
| *Asian/Asian British* | 72 | 0.6 | 36 | 0.4 | 29 | 1.6 | 7 | 3.9 |
| *Mixed race* | 43 | 0.4 | 33 | 0.3 | 8 | 0.4 | 2 | 1.1 |
| *Other or unsure* | 94 | 0.8 | 66 | 0.6 | 26 | 1.4 | 2 | 1.1 |
| *Missing* | 1736 | | 1284 | | 399 | | 53 | |
| ***Social class at birth*** | N = 14094 | | N = 11625 | | N = 2234 | | N = 235 | |
| *I* | 535 | 4.2 | 522 | 4.9 | 12 | 0.6 | 1 | 0.5 |
| *II* | 1652 | 13.0 | 1557 | 14.7 | 85 | 4.4 | 10 | 5.3 |
| *III non-manual* | 1272 | 10.0 | 1142 | 10.8 | 117 | 6.1 | 13 | 6.9 |
| *III manual* | 6474 | 50.9 | 5367 | 50.6 | 1029 | 53.2 | 78 | 41.3 |
| *IV* | 1571 | 12.3 | 1207 | 11.4 | 335 | 17.3 | 29 | 15.3 |
| *V* | 1228 | 9.6 | 815 | 7.7 | 355 | 18.4 | 58 | 30.7 |
| *Missing* | 1362 | | 1015 | | 301 | | 46 | |
| **Legal Marital status at age 44** | N = 14094 | | N = 11625 | | N = 2234 | | N = 235 | |
| *Single/Never married* | 834 | 10.7 | 703 | 10.3 | 110 | 12.0 | 21 | 39.6 |
| *Married* | 5637 | 72.1 | 5010 | 73.1 | 603 | 65.9 | 24 | 45.3 |
| *Separated/divorced/widowed* | 1347 | 17.2 | 1137 | 16.6 | 202 | 22.1 | 8 | 15.1 |
| *Missing* | 6276 | | 4775 | | 1319 | | 182 | |
| **Self-reported general health age 44** | N = 14094 | | N = 11625 | | N = 2234 | | N = 235 | |
| *Excellent* | 1344 | 17.2 | 1241 | 18.1 | 94 | 10.3 | 9 | 17.0 |
| *Good* | 5000 | 63.9 | 4432 | 64.6 | 547 | 59.7 | 21 | 39.6 |
| *Fair* | 1322 | 16.9 | 1061 | 15.5 | 240 | 26.2 | 21 | 39.6 |
| *Poor* | 160 | 2.0 | 122 | 1.8 | 36 | 3.9 | 2 | 3.8 |
| *Missing* | 6268 | | 4769 | | 1317 | | 182 | |

Note

[a] N = number of observations

[b] SD = standard deviation

c synthesised IQ (Intelligence Quotient) scores are calculated by conducting a PCA of the four cognitive tests for participants who completed all cognitive tests at age 11 (N = 14,094), to obtain component scores, then transformed the resulting scores to a scale with a mean of 100 (SD = 15) to correspond to IQ scores (min = 50.1, max = 139.6) and these scores were normally distributed.

## Association between IQ classifications and social participation

As shown in Table 2, in the fully adjusted model (model 4), people with mild intellectual disability had a significantly higher social network index (1.2, p = 0.03, [0.1,2.2]) and social contact with relatives at age 44 (0.8, p = 0.004, [0.3,1.4]) than people with IQ above 85.

People with borderline intellectual functioning had significantly higher social contact with relatives compared with people with IQ above 85 (0.2, p = 0.011, [0.0,0.3]), but less social contact with friends (-0.2, p = 0.038, [-0.3,0]) and less confiding/emotional support from the closest person (-0.4, p = 0.017, [-0.6,0.1]) at age 44 and fewer confiding relationships with anyone at age 50 (-0.2, p<0.001, [-0.2,-0.1]).

## Mediation analysis: Social participation mediation of the relationships between IQ in childhood and cognition, and QoL at age 50

Table 3 shows the mediation analysis results for social participation in relation to the associations between IQ in childhood and cognition at age 50 for the three groups. For people with borderline intellectual functioning, IQ in childhood predicted cognition at age 50 and was only related to their confiding relationships with anyone (6.4%), but not to other aspects of social participation. But for people with IQ above 85, the effect of IQ on their cognition at age

**Table 2. Comparison of social participation between people with borderline and mild intellectual disability and people with IQ above 85 linear regression model (MICE (Multiple Imputation by Chained Equations)).**

| | | Model 1[a] (unadjusted) | | Model 2[b] (adjusted) | | Model 3[c] (adjusted) | | Model 4[d] (adjusted) | |
|---|---|---|---|---|---|---|---|---|---|
| | | Regression coefficient [95% C.I.] | P value | Regression coefficient [95% C.I.] | P value | Regression coefficient [95% C.I.] | P value | Regression coefficient [95% C.I.] | P value |
| Social network index (age 44) (n = 5651) | IQ above 85 | reference | | Reference | | reference | | reference | |
| | Borderline intellectual functioning | 0.0 (-0.2,0.3) | 0.928 | 0.0 (-0.3,0.2) | 0.888 | 0.0 (-0.3,0.3) | 0.989 | 0.0 (-0.2,0.3) | 0.742 |
| | Mild intellectual disability | 1.0 (-0.1,2.0) | 0.082 | 0.9 (-0.1,2.0) | 0.092 | 1.0(-0.1,2.0) | 0.071 | 1.2 (0.1,2.2) | 0.030 |
| Social contact index- relatives (age 44) (n = 6182) | IQ above 85 | reference | | Reference | | reference | | reference | |
| | Borderline intellectual functioning | 0.2 (0.1,0.4) | 0.002 | 0.2 (0.0,0.3) | 0.025 | 0.2 (0.0,0.3) | 0.017 | 0.2 (>0.0,0.3) | 0.011 |
| | Mild intellectual disability | 0.7 (0.2,1.3) | 0.012 | 0.7 (0.1,1.3) | 0.018 | 0.8 (0.2,1.4) | 0.006 | 0.8 (0.3,1.4) | 0.004 |
| Social contact index- friends (age 44) (n = 6650) | IQ above 85 | reference | | Reference | | reference | | reference | |
| | Borderline intellectual functioning | -0.2 (-0.4, -0.1) | 0.003 | -0.2 (-0.4,0) | 0.012 | -0.2 (-0.4,0.0) | 0.018 | -0.2 (-0.3, <0.0) | 0.038 |
| | Mild intellectual disability | 0.3 (-0.4,0.9) | 0.413 | 0.3 (-0.4,0.9) | 0.416 | 0.3 (-0.4,0.9) | 0.426 | 0.3 (-0.3,1.0) | 0.326 |
| confiding/emotional support from the closest person (age 44) (n = 7202) | IQ above 85 | reference | | Reference | | reference | | reference | |
| | Borderline intellectual functioning | -0.6 (-0.9, -0.3) | <0.001 | -0.5 (-0.8, -0.2) | 0.001 | -0.4 (-0.7, -0.1) | 0.014 | -0.4 (-0.6, -0.1) | 0.017 |
| | Mild intellectual disability | -0.3 (-1.5,0.9) | 0.629 | -0.2 (-1.4,1.0) | 0.776 | 0.5 (-0.7,1.7) | 0.403 | 0.5 (-0.7,1.7) | 0.381 |
| Confiding relationships with anyone (age 50) (n = 8385 | IQ above 85 | reference | | reference | | reference | | reference | |
| | Borderline intellectual functioning | -0.2 (-0.2, -0.1) | <0.001 | -0.2 (-0.2, -0.1) | <0.001 | -0.2 (-0.2, -0.1) | <0.001 | -0.2 (-0.2, -0.1) | <0.001 |
| | Mild intellectual disability | -0.2 (-0.3,0.0) | 0.026 | -0.2 (-0.3,0) | 0.036 | -0.1 (-0.3,0) | 0.119 | -0.1 (-0.3, >0.0) | 0.165 |

Notes

[a] Model 1 = unadjusted

[b] Model 2 = adjusted for sex, ethnic group, and social class of mother's husband at birth

[c] Model 3 = adjusted additionally for legal marital status at age 44 and self-reported general health at age 44; Both figures indicate p<0.05 in multivariable analysis.

[d] Model 4 = adjusted additionally for continuous total CIS-R scores at age 44

[e] C.I. = Confidence interval.

[f] N = number of observations, it is the number of participants in each model, including people with IQ above 85, people with borderline intellectual functioning and people with mild intellectual disability. [g] Regression coefficients are the regression coefficients capturing the relationships between the focal variables in each model.

**Table 3. The mediation effect of social participation on the relationship between IQ at age 11 and cognition or quality of life at age 50.**

| | | Relationship between IQ and cognition | | | Relationship between IQ and QoL | | |
|---|---|---|---|---|---|---|---|
| | | IQ above 85 | Borderline intellectual functioning | Mild intellectual disability | IQ above 85 | Borderline intellectual functioning | Mild intellectual disability |
| *Social network* | N | 4322 | 548 | 23 | 4076 | 502 | 17 |
| | Mediation effect | No mediation | No mediation | No mediation | No mediation | No mediation | No mediation |
| *- with relatives* | N | 4715 | 601 | 29 | 4450 | 548 | 22 |
| | Mediation effect | 1.2, partial mediation | No mediation | No mediation | -12.5, partial mediation | No mediation | No mediation |
| *- with friends* | N | 5104 | 610 | 26 | 4825 | 555 | 19 |
| | Mediation effect | No mediation | No mediation | No mediation | 14.4, partial mediation | No mediation | No mediation |
| *Confiding/emotional support* | N | 5537 | 653 | 30 | 5243 | 593 | 23 |
| | Mediation effect | No mediation | No mediation | No mediation | No mediation | No mediation | No mediation |
| *Confiding relationships with anyone* | N | 7071 | 973 | 57 | 6550 | 850 | 46 |
| | Mediation effect | 0.5, partial mediation | 6.4, partial mediation | No mediation | 13.4, partial mediation | 27.4, partial mediation | No mediation |

50 was partially mediated by social contact with relatives (1.2%) and confiding relationships with anyone (0.5%) (S1 Table). For people with borderline intellectual functioning, confiding relationships with anyone (27.4%) had a partially mediating effect on the relationship between IQ in childhood and QoL at age 50. For people with IQ above 85, the association between IQ in childhood and QoL at age 50 was partially mediated by social contact with friends (14.4%), and confiding relationships with anyone (13.4%), and partially negatively mediated by social contact with relatives (-12.5%), which means social contact with relatives strengthened the association between IQ and QoL at age 50 (S2 Table).

## Discussion

This study of a birth cohort followed for over 50 years with detailed cognitive testing in childhood has identified that people with mild intellectual disability had more social contact with their family in midlife; and that people with borderline intellectual functioning had more social contact with family, but less with friends and fewer confiding relationships than those with intellectual functioning in the normal range in midlife. Even at the age of 50, when dementia would be rare, confiding relationships mediated better cognition for people with borderline intellectual functioning and were important for QoL.

### Social network with relatives and friends

We found that people with mild intellectual disability had a higher social contact index than other groups because of more social contact with their relatives. This may be linked to their greater dependency and reduced ability to live independently during midlife. However, they had lower social contact with friends. Also, we found people with borderline intellectual functioning had less social contact with friends, less strong confiding/emotional support from the closest person, and less confiding relationships with anyone, but more social contact with relatives than people of IQ above 85.

Besides extending our knowledge, those findings indicate that people with borderline intellectual functioning or mild intellectual disability may have considerable social contact with

relatives but have challenges in building close social relationships with people outside their family. It is consistent with a study from the Irish Longitudinal Study on Ageing, which found only just over half of respondents with intellectual disability said they had friends outside home [8].

## The closest person

Our analysis of social participation found that, compared to people with IQ above 85 and those with borderline intellectual functioning, individuals with mild intellectual disability reported greater contact with their partner and parents but fewer with boyfriends/girlfriends, children, and other relatives. An Irish study reported that most people with intellectual disability are not in intimate relationships; 99% of older people of all severity of intellectual disability are single [29].

Only 5.6% of people with mild intellectual disability and 6.9% of people with borderline intellectual functioning reported their closest persons were friends, compared to 8.6% of people with IQ above 85, most people regardless of their IQ reported that their closest person was a relative. A study with 753 adults with intellectual disability found that only around half of the participants had a best friend, with options including another person with intellectual disability (63.7%), a staff member (15.8%), a family member (8.6%) and other (6.8%) [4]. Another study reported that people with intellectual disability feel more comfortable and able to form 'true friendships' with others who share their identity [30]. However, some people judged those friendships between people with and without intellectual disability maybe an indicator of successful social participation [31, 32].

It is noteworthy that, compared with people with borderline intellectual functioning (0.3%) and IQ above 85 (0.4%), 8.3% of people with mild intellectual disability reported their closest persons are 'others', possibly because the questionnaire did not include "staff or carers" as a possible option.

## Findings for midlife cognition

All types of social participation were significantly associated with cognition at age 50 for people with borderline intellectual functioning and IQ above 85, but for people with mild intellectual disability only confiding/emotional support and confiding relationships were significantly associated.

This birth cohort also allowed us to test the hypothesis that social participation mediated the association between IQ and cognition. We found that confiding relationships were potentially important for the future cognition of people with borderline intellectual functioning, because they mediated 6.4% of the effect of IQ in childhood on cognition at age 50, compared to the smaller (0.5%) mediating effect for people with IQ above 85, although we were unable to test this in people with mild intellectual disability due to the small sample size. Also, since the analysis of the association between confiding relationships with anyone and midlife cognition is cross-sectional, it is possible that people with better cognition have more confiding relationships with anyone. A potential explanation for the difference in the association for confiding relationships between people with borderline intellectual functioning and IQ above 85 is that we examined cognitive scores at age 50, an age at which there is little cognitive decline in the general population compared to those with borderline intellectual functioning. Therefore, the potential mediating effect of confiding relationships on cognition may manifest earlier for people with borderline intellectual functioning than for those without intellectual disability. This finding is similar to a study using data from the national Midlife in the U.S. study (MIDUS) [33], which used the Brief Test of Adult Cognition by Telephone (BTACT) [34] to measure

cognitive functioning. They found that social engagement is related to cognition and that this association is stronger in people aged under 65.

## Findings about QoL

Adults with intellectual disability had worse QoL at age 50 than people with IQ above 85. Social contact with friends, confiding/emotional support and confiding relationships are significantly related to better QoL for people with borderline intellectual functioning. However, for people with IQ above 85, all types of social participation were significantly related to a higher QoL at age 50. A study using the Irish Longitudinal Study on Ageing (TILDA) data found that being involved in various forms of social participation, such as intimate social relationships, formal activities outside of work, active and social leisure, as well as passive and solitary leisure, was linked to the higher quality of life [29].

Findings in this study suggested that building confiding relationships can help reduce loneliness and benefit QoL for people with intellectual disability. In this study, we found confiding relationships at age 50 mediated 27.4% of the relationship between IQ and QoL for people with borderline intellectual functioning. However, it is uncertain why confiding relationships with anyone at age 50 have a mediating effect on the relationships between IQ in childhood and cognition or QoL at age 50, but the confiding/emotional support from the closest person (age 44) does not. A potential explanation is that, for people with borderline intellectual functioning, confiding relationships with anyone are particularly important sources of social support and relevant for cognition [35], and the number of confidants can be more than one. Also, adults with intellectual disability may identify staff as their confidants because staff often provide them with primary social companions and social resources [29, 36], but staff might not be considered as their closest persons at age 44. The implication is that if adults with borderline intellectual functioning had confiding relationships with more than one person, someone else who was not necessarily the closest person but outside their families might be able to act as a confidant and this would extend the size of their social networks and thus might confer protection from cognition decline.

**Strengths and limitations.** This is the first study to use a nationally representative sample to investigate whether confiding relationships with others can protect cognition for people with borderline intellectual function and people with IQ above 85. This study also provides knowledge about the structures of social participation in a representative cohort of adults with borderline intellectual functioning and mild intellectual disability, in comparison to that of people with IQ above 85. To maintain representativeness, we imputed values of missing data using MICE.

This study has limitations. We were unable to determine whether the confidants of adults with intellectual disability are individuals themselves with or without intellectual disability, or to compare the difference in mediation effect of social participation between people with different severity of intellectual disability due to the lack of information and power. We only considered social contact with relatives and friends, and confiding relationships but did not include group activities or work, so we may have underestimated social participation, particularly for people with borderline intellectual functioning as their patterns of social participation may be different from people without intellectual disability. Also, people with mild intellectual disability who said they had a partner or boyfriend/girlfriend might understand the question not as an exclusive, intimate partner but as someone special to support them, as the number of partners reported was much larger than previously reported in people with mild intellectual disability.

Furthermore, for people with mild intellectual disability, the available data did not fulfil the criteria of Baron and Kenny's Mediation test, so mediation is not calculated. This may be due

to the small samples or because there is no mediation effect. We were unable to examine whether the apparent effect of social participation on cognition and QoL will continue as these people become older. And since the incidence of dementia is low at age 50, we were unable to explore whether social participation in midlife affects dementia risk. We need to explore it in the future.

## Conclusion

The findings in this study indicate that adults with borderline intellectual functioning and mild intellectual disability often have positive relationships within their family but less so outside their home environment. In addition, their social relationships may not provide the same level of participation and opportunities for adaptation to new social situations that come with interacting with a wider community. This study also provides initial evidence that, although there was little cognitive decline by age 50, confiding relationships still had a mediating role for cognitive function in people with borderline intellectual functioning and can contribute to a better midlife QoL for people with IQ above 85 as well as people with borderline intellectual functioning. This informs future clinical research into interventions for people with intellectual disability as it provides evidence that social participation may be particularly important to protect cognition in this vulnerable group. Furthermore, social participation among adults with intellectual disability is likely to be impacted by societal attitudes, which may change over time. As such, it is important to further investigate how to facilitate social participation for this population.

## Supporting information

**S1 Appendix. Questionnaires.**
(DOCX)

**S2 Appendix. Baron and Kenny method and statistical significance tests techniques.**
(DOCX)

**S1 Table. The mediation effect of social participation on the relationship between IQ at age 11 and cognition at age 50.**
(DOCX)

**S2 Table. The mediation effect of social participation on the relationship between IQ at age 11 and QoL at age 50.**
(DOCX)

**S3 Table. IQ test scores, cognitive test scores and social participation in study participants.**
(DOCX)

## Author Contributions

**Conceptualization:** Zuyu Wang, Andrew Sommerlad, Angela Hassiotis, Gill Livingston.

**Data curation:** Zuyu Wang.

**Formal analysis:** Zuyu Wang.

**Investigation:** Zuyu Wang.

**Methodology:** Zuyu Wang, Gill Livingston.

**Project administration:** Zuyu Wang.

**Resources:** Zuyu Wang.

**Software:** Zuyu Wang.

**Supervision:** Andrew Sommerlad, Angela Hassiotis, Gill Livingston.

**Writing – original draft:** Zuyu Wang.

**Writing – review & editing:** Andrew Sommerlad, Angela Hassiotis, Marcus Richards, Gill Livingston.

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
