## [Decision Letter · Decision Letter 0]

28 Feb 2024

PONE-D-23-29741Mid-life social participation in people with intellectual disability: the 1958 British birth cohort studyPLOS ONE

Dear Dr. Wang,

Thank you for submitting your manuscript to PLOS ONE. After careful consideration, we feel that it has merit but does not fully meet PLOS ONE’s publication criteria as it currently stands. Therefore, we invite you to submit a revised version of the manuscript that addresses the points raised during the review process.

We look forward to receiving your revised manuscript.

Kind regards,

Robert Didden

Academic Editor

PLOS ONE

Journal Requirements:

3. Please amend the manuscript submission data (via Edit Submission) to include author Andrew Sommerlada, Angela Hassiotisa, Marcus Richards and Gill Livingston.

4. Please include your tables as part of your main manuscript and remove the individual files. Please note that supplementary tables (should remain/ be uploaded) as separate "supporting information" files

Additional Editor Comments:

Please find below the detailed comments of Reviewer #1. The paper is of interest but may benefit from a revision. The comments are straightforward and largely related to issues of clarification. Also pay attention to the reviewer's comment regarding clinical implications of this study.

Reviewers' comments:

Reviewer's Responses to Questions

**Comments to the Author**

1. Is the manuscript technically sound, and do the data support the conclusions?

Reviewer #1: Partly

2. Has the statistical analysis been performed appropriately and rigorously? 

Reviewer #1: I Don't Know

3. Have the authors made all data underlying the findings in their manuscript fully available?

Reviewer #1: Yes

4. Is the manuscript presented in an intelligible fashion and written in standard English?

Reviewer #1: Yes

5. Review Comments to the Author

Reviewer #1: The article “Mid-life social participation in people with intellectual disability: the 1958 British birth cohort study ”is based on the 1958-birth cohort in England, Scotland and Wales, followed-up at ages 44 and 50 years. The study targets different aspects of social participation, and QoL and compares cognitive subgroups; those with mild intellectual disability, borderline intellectual functioning and those with IQs above 85.

The article is well written and mostly clear.

Abstract:

I wonder about this sentence: “Having a confiding relationship partially mediated the association at age 50 between IQ and cognition (6.4%) and QoL (27.4%) for people with borderline intellectual functioning”. Is the result one of the main findings?

Background:

Page 4, regarding study aims: “2) Does social participation (social contacts and confiding relationships) mediate the association between intelligence quotient scores (IQ) in childhood and future cognition and QoL in people with and without intellectual disability?” My question is if it could also be the opposite, i.e., that IQ in childhood mediates later social participation?

Demographic information:

It would be valuable if the authors explained RG I, II, III

IQ in childhood:

I wonder about the test methods at age 11 years, were there test data from some established tests, such as the WISC? Perhaps this is explained in reference no 15.

Were tests performed by psychologists?

The mild intellectual disability range is broad, IQ appr. 50- appr.70. Was that considered?

The three different parts dealing with Social participation:

My question is if the participants with mild intellectual disability answered these forms entirely by themselves or if they received some help, perhaps explaining a question?

Measurement of cognitive ability in midlife:

I have a similar question here about the test data, was there any correlation with the WAIS test? Who carried out the test procedure? Psychologists?

Do the authors know if most of those who participated had results in the IQ “higher” IQ range 60-70?

Health status:

It is very valuable that also mental health status is included. However, I have the same question here as previously, did the participants with mild intellectual disability receive any help to understand the questions.

Results:

Social network:

First line, I think it is important to specify that it is mild intellectual disability

Mediation analysis: "Social participation mediation of the relationships between IQ in childhood and cognition, and QoL at age 50": To me, this part is not so easy to understand. However, the Tables in the Appendix are explanatory.

My question: When the authors mention cognition in the adults, does it include an IQ measure? Were IQ levels within the same areas in adulthood as at age 11?

Discussion:

The closest person:

Please check if intellectual disability is the correct term, or if it should be mild intellectual disability. Perhaps it would be more relevant if the group consisted of those with mild intellectual disability.

“A study reported that most people with intellectual disability are not in intimate relationships; 99% of older people with intellectual disability are single(28).” I think the authors should add if the percentage 99% refers to all levels of intellectual disability.

The sentence:” Only 5.6% of people with mild intellectual disability and 6.9% of people with borderline intellectual functioning reported their closest persons were friends, compared to 8.6% of people with IQ above 85”. To me the percentages seem very similar, maybe that should be commented on.

Findings for midlife cognition:

The sentences: “This birth cohort also allowed us to test the hypothesis that social participation mediated the association between IQ and cognition.”

My question is if it is not exactly the opposite, i.e., that that IQ and cognition were of importance for social participation.

The following part seems for me a bit too detailed, is it important information? “We found that confiding relationships were potentially important for the future cognition of people with borderline intellectual functioning, because they mediated 6.4% of the effect of IQ in childhood on cognition at age 50, compared to the smaller (0.5%) mediating effect for people with IQ above 85, although we were unable to test this in people with mild intellectual disability due to the small sample size.”

Findings about QoL:

To me it is difficult to understand the relevance of the following sentence: “An implication is that if adults with borderline intellectual functioning had confiding relationships with more than one person, someone else who was not necessarily the closest person but outside their families might be able to act as a confidant to extend the size of their social networks and thus confer protection from cognition decline.”

Finally, I think the authors should give more clinical implications of their study. How will the study populations and professionals benefit from this study and its results?

6. PLOS authors have the option to publish the peer review history of their article (what does this mean?). If published, this will include your full peer review and any attached files.

Reviewer #1: No

---

## [Author Response · Author response to Decision Letter 0]

25 Mar 2024

Mid-life social participation in people with intellectual disability: the 1958 British birth cohort study

Thank you for the comments from editors and reviewers. We have addressed these point by point below with our response in blue and revised sections of the manuscript in italics. 

Reviewer #1. 

1. Reviewer #1: The article “Mid-life social participation in people with intellectual disability: the 1958 British birth cohort study” is based on the 1958-birth cohort in England, Scotland and Wales, followed-up at ages 44 and 50 years. The study targets different aspects of social participation, and QoL and compares cognitive subgroups; those with mild intellectual disability, borderline intellectual functioning and those with IQs above 85.

The article is well written and mostly clear. 

Response: Thank you very much.

2. Abstract:

I wonder about this sentence: “Having a confiding relationship partially mediated the association at age 50 between IQ and cognition (6.4%) and QoL (27.4%) for people with borderline intellectual functioning”. Is the result one of the main findings?

Response: Yes, it is one of the main findings, and helps to answer our research question (2).

3. Background:

Page 4, regarding study aims: “2) Does social participation (social contacts and confiding relationships) mediate the association between intelligence quotient scores (IQ) in childhood and future cognition and QoL in people with and without intellectual disability?” My question is if it could also be the opposite, i.e., that IQ in childhood mediates later social participation?

Response: We agree with the reviewer, IQ is associated with later social participation, and that is the first research question in this study, and we show the analysis in the results. In the second research question, we investigated how midlife social participation has impact on the relationship between IQ in childhood and cognition in later life.

4. Demographic information:

It would be valuable if the authors explained RG I, II, III

Response: Thank you for pointing this out. We have rewritten the sentence as ‘social class was defined as the employment of the mother’s partner at birth using The Registrar-General’s Social Class Scheme: The Stevenson Version. (RG I, II, III non-manual, III manual, IV, V); ’ , and added a reference: Carr-Hill RA, Pritchard CW, Carr-Hill RA, Pritchard CW. The registrar-general’s social class scheme: The Stevenson version. Women’s Social Standing: The Empirical Problem of Female Social Class. 1992;10–24. 

5. IQ in childhood:

I wonder about the test methods at age 11 years, were there test data from some established tests, such as the WISC? Perhaps this is explained in reference no 15.

Were tests performed by psychologists?

Response: Yes, it is explained in the reference no.16 (was no.15). As specified in the methods, at age 11 the tests used to calculate IQ covered (1) general ability, (2) reading comprehension, (3) mathematical tests and (4) copying design. Tests 1,2 and 4 were prepared by the National Foundation for Educational Research in England and Wales. The tests were administered by teachers. 

The mild intellectual disability range is broad, IQ appr. 50- appr.70. Was that considered?

Responses: Yes, we are using the definition of mild intellectual disability according to ICD-10: 2019, F70 Mild mental retardation: approximate IQ range of 50 to 69 (in adults, mental age from 9 to under 12 years).

6. The three different parts dealing with Social participation:

My question is if the participants with mild intellectual disability answered these forms entirely by themselves or if they received some help, perhaps explaining a question?

Response: There was an interviewer or nurse present who could help if needed. At age 44, the questions about social participation are included in the self-completion booklet. Participants were asked to complete it themselves before a nurse came to see them, then they gave it to the nurse during the visit. Participants were able to ask the nurse for help with the questionnaire. At age 50, the questionnaire was not included in the self-completion questionnaire but was read out by the interviewer using a show card presented by the interviewer. Thus participants could ask help from the interviewer immediately if they could not understand the question.

7. Measurement of cognitive ability in midlife:

I have a similar question here about the test data, was there any correlation with the WAIS test? Who carried out the test procedure? Psychologists?

Response: The tests were administered by trained interviewers working on these surveys. The WAIS was not administered. All tests were read out or performed by trained interviewers, working on surveys including the English Longitudinal Study of Ageing (ELSA) (http://www.ifs.org.uk/elsa/)

8. Do the authors know if most of those who participated had results in the IQ “higher” IQ range 60-70?

Response: Yes, we know the distribution of IQ scores. Please see the graph below, the x axis is the IQ scores of participants with mild intellectual disability in this study and more of them had IQ of 60 to 70.

9. Health status:

It is very valuable that also mental health status is included. However, I have the same question here as previously, did the participants with mild intellectual disability receive any help to understand the questions.

Response: The CIS-R questionnaires was computer administered and self-completed, with the support of trained nurses. Health questions were included in the self-completion booklet, and participants were asked to complete and return this with their biomedical samples. If participants had any questions or needed any help, they could ask from nurses or research team.

10. Results:

Social network:

First line, I think it is important to specify that it is mild intellectual disability

Response: Thank you for pointing this out. We have added the word “mild” 

11. Mediation analysis: "Social participation mediation of the relationships between IQ in childhood and cognition, and QoL at age 50": To me, this part is not so easy to understand. However, the Tables in the Appendix are explanatory.

Response: Thank you for your comment and saying it was explained. and we have ensured that it is clear. 

12. My question: When the authors mention cognition in the adults, does it include an IQ measure? Were IQ levels within the same areas in adulthood as at age 11?

Response: As specified in the methods, at age 11 the tests used to calculate IQ covered general ability, reading comprehension, mathematical tests and copying. The measures in adulthood tested memory, verbal fluency and speed of processing. These latter tests are common cognitive tests used in adulthood but are not intended to provide an IQ measure.

13. Discussion:

The closest person:

Please check if intellectual disability is the correct term, or if it should be mild intellectual disability. Perhaps it would be more relevant if the group consisted of those with mild intellectual disability.

“A study reported that most people with intellectual disability are not in intimate relationships; 99% of older people with intellectual disability are single(28).” I think the authors should add if the percentage 99% refers to all levels of intellectual disability.

Response: It is about all levels of intellectual disability, and we have added ‘all severity’ in the sentence. As it covers mild intellectual disability, and the finding is that nearly everyone is single we think it is relevant. 

14. The sentence:” Only 5.6% of people with mild intellectual disability and 6.9% of people with borderline intellectual functioning reported their closest persons were friends, compared to 8.6% of people with IQ above 85”. To me the percentages seem very similar, maybe that should be commented on.

Response: Thank you. We have added “Most people regardless of their IQ reported that their closest person was a relative”

Findings for midlife cognition:

15. The sentences: “This birth cohort also allowed us to test the hypothesis that social participation mediated the association between IQ and cognition.”

My question is if it is not exactly the opposite, i.e., that that IQ and cognition were of importance for social participation.

Response: We discussed this above- see answer to question 3.

16. The following part seems for me a bit too detailed, is it important information? “We found that confiding relationships were potentially important for the future cognition of people with borderline intellectual functioning, because they mediated 6.4% of the effect of IQ in childhood on cognition at age 50, compared to the smaller (0.5%) mediating effect for people with IQ above 85, although we were unable to test this in people with mild intellectual disability due to the small sample size.”

Response: Thank you for your comment. We think it is important and suggests that people with ID show effects of isolation on cognition at an earlier age than people with higher IQ.

17. Findings about QoL:

To me it is difficult to understand the relevance of the following sentence: “An implication is that if adults with borderline intellectual functioning had confiding relationships with more than one person, someone else who was not necessarily the closest person but outside their families might be able to act as a confidant to extend the size of their social networks and thus confer protection from cognition decline.”

Response: We are sorry this was not clear and have rewritten this to say “ The implication is that if adults with borderline intellectual functioning had confiding relationships with more than one person, someone else who was not necessarily the closest person but outside their families might be able to act as a confidant and this would extend their social participation and thus might confer protection from cognition decline.”

18. Finally, I think the authors should give more clinical implications of their study. How will the study populations and professionals benefit from this study and its results?

Response: Thank you for your comments, we agree with the reviewer. We have added “ This informs future clinical research into interventions for people with intellectual disability as it provides evidence that social participation may be particularly important to protect cognition in this vulnerable group.”

---

## [Decision Letter · Decision Letter 1]

3 Apr 2024

Mid-life social participation in people with intellectual disability: the 1958 British birth cohort study

PONE-D-23-29741R1

Dear Dr. Wang,

We’re pleased to inform you that your manuscript has been judged scientifically suitable for publication and will be formally accepted for publication once it meets all outstanding technical requirements.

Kind regards,

Robert Didden

Academic Editor

PLOS ONE

Additional Editor Comments (optional):

Reviewers' comments:

Reviewer's Responses to Questions

**Comments to the Author**

1. If the authors have adequately addressed your comments raised in a previous round of review and you feel that this manuscript is now acceptable for publication, you may indicate that here to bypass the “Comments to the Author” section, enter your conflict of interest statement in the “Confidential to Editor” section, and submit your "Accept" recommendation.

Reviewer #1: All comments have been addressed

2. Is the manuscript technically sound, and do the data support the conclusions?

Reviewer #1: Yes

3. Has the statistical analysis been performed appropriately and rigorously? 

Reviewer #1: I Don't Know

4. Have the authors made all data underlying the findings in their manuscript fully available?

Reviewer #1: Yes

5. Is the manuscript presented in an intelligible fashion and written in standard English?

Reviewer #1: Yes

6. Review Comments to the Author

Reviewer #1: Thank you very much for your clear answers to my questions!

I would just like to suggest that it could be mentioned that it was teachers who carried out the cognitive tests in chikdhood. Perhaps it could also be mentioned if the teachers had any collaboration with a psychologist with regard to interpretation of test results.

7. PLOS authors have the option to publish the peer review history of their article (what does this mean?). If published, this will include your full peer review and any attached files.

Reviewer #1: No

---

## [Editor Report · Acceptance letter]

7 May 2024

PONE-D-23-29741R1 

PLOS ONE

Dear Dr. Wang, 

I'm pleased to inform you that your manuscript has been deemed suitable for publication in PLOS ONE. Congratulations! Your manuscript is now being handed over to our production team.

Kind regards, 

on behalf of

Professor Robert Didden 

Academic Editor

PLOS ONE